# Applications for Circulating Cell-Free DNA in Oral Squamous Cell Carcinoma: A Non-Invasive Approach for Detecting Structural Variants, Fusions, and Oncoviruses

**DOI:** 10.3390/cancers17121901

**Published:** 2025-06-06

**Authors:** Mahua Bhattacharya, Dan Yaniv, Dylan P. D’Souza, Eyal Yosefof, Sharon Tzelnick, Rajesh Detroja, Tal Wax, Adva Levy-Barda, Gideon Baum, Aviram Mizrachi, Gideon Bachar, Milana Frenkel Morgenstern

**Affiliations:** 1Azrieli Faculty of Medicine, Bar Ilan University, Safed 1311502, Israel or bmahua@yahoo.com (M.B.); dylan.dsouza@biu.ac.il (D.P.D.); rajesh.detroja@biu.ac.il (R.D.); eliezer.baum@biu.ac.il (G.B.); 2Otolaryngology, Head and Neck Surgery Department, Rabin Medical Center, Petah Tikva 4941492, Israel; danya1@clalit.org.il (D.Y.); eyalyo@clalit.org.il (E.Y.); aviramguy@hotmail.com (A.M.); 3Scojen Institute of Synthetic Biology, The Dina Recanati School of Medicine, Reichman University, Herzliya 4610101, Israel; 4Division of Head and Neck Surgery, Princess Margaret Cancer Center, University of Toronto, Toronto, ON M5G 2M9, Canada; tzelnicksharon@gmail.com; 5Princess Margaret Cancer Centre, University Health Network, Toronto, ON M5G 2M9, Canada; 6Department of Pathology, Rabin Medical Centre, Petah Tikva 4941492, Israel; ruhamaa@clalit.org.il; 7Biobank, Department of Pathology, Rabin Medical Centre, Petah Tikva 4941492, Israel; advale11@clalit.org.il; 8Department of Digital Medical Technologies, Holon Institute of Technology, Holon 5883754, Israel

**Keywords:** liquid biopsy, cfDNA, oral cavity cancers, OSCC, gene fusion, CNVs, SNVs, virus integration

## Abstract

A comprehensive study analyzed circulating cell-free DNA (cfDNA) from 30 oral squamous cell carcinoma (OSCC) patients, 199 public OSCC, and 192 normal samples to explore its potential in early-stage diagnostics and prognosis. The findings suggest that the cfDNA concentration correlates with the tumor stage, malignancy, and survival prognosis in OSCC patients. Deep genome sequencing of cfDNA revealed several genomic alterations: 1. Copy number variation analysis identified amplifications and deletions at loci 1q, 2q, 3p, 3q, and 8q22. Notably, amplifications of TP53 and PIK3CA, along with other genes related to keratinization, were identified in relapsed OSCC cases. Single nucleotide variations were observed in genes highly mutated in OSCC. 2. A novel fusion gene, TRMO-TRNT1, was detected in seven high-grade tumor samples. The parental genes, TRMO and TRNT1, are involved in tRNA modification and DNA repair, respectively. 3. Integrations of human papillomavirus, simian virus, and enterovirus were found in the OSCC samples, potentially indicating viral involvement in OSCC pathogenesis. These results underscore the utility of cfDNA as a non-invasive biomarker for OSCC, offering insights into tumor genomics and aiding in early detection and prognosis assessment.

## 1. Introduction

Oral squamous cell carcinoma (OSCC) considered a part of head and neck squamous cell carcinoma (HNSCC) [1], affects the oral and buccal regions, including the gums, the floor of the mouth, and the tongue, palate, and lips [2]. OSCC is the 16th most common form of cancer and the 15th highest contributor to morbidity rates due to cancer [3]. Oral cancer is mostly associated with lifestyle, being linked to smoking and drinking, although some authors suggest that a predisposition to human papillomavirus (HPV) is also linked to OSCC [3,4,5,6]. While early diagnosis can help prevent metastasis and progression, OSCC often goes undiagnosed until a late stage when symptoms are observed, making radiotherapy and chemotherapy less effective [7].

The discovery of non-invasive methods, like a liquid biopsy, has led to progress in cancer diagnostics [8,9,10,11], specifically improving the diagnosis of lung, blood, colorectal, and cervical cancers, among others. At the same time, a liquid biopsy analysis of cell-free DNA (cfDNA) in plasma, saliva, and urine has helped determine the survival prognosis of patients [8,9,12,13,14,15]. CfDNA fragments appear in the plasma because of cellular apoptosis or necrosis. The amount of released cfDNA correlates with the tumor burden and metastasis [12,16,17,18,19,20,21]. These approaches offer rapid, repeatable sampling with minimal risk to the patient. As technology advances, liquid biopsy is increasingly being explored for early detection, treatment monitoring, and recurrence prediction across a broader spectrum of malignancies [8,9,12,13,22]. The cfDNA analyses have already been conducted on HNSCC patients, specifically on OSCC patients [23,24]. Human papillomavirus (HPV) is one of the major causes of HNSCC, with a significantly mutated *HPV* gene being the major cause of HNSCC [25,26,27]. The presence of HPV and other pathogens, as revealed by cfDNA analyses of OSCC patients, has made diagnosis and prognosis at the early stages possible [1,27,28,29].

Previous studies reported the presence of a second virus, herpes simplex virus (HSV), in HNSCC and other squamous cell carcinomas, like cervical cancer, and correlated the presence of HPV and HSV with the oncogenesis of these cancers [30,31,32]. Since OSCC is one of the most common head and neck cancers, we were interested in analyzing the cfDNA from patients belonging to an Israeli cohort to identify the genomic alterations associated with the disease, such as the copy number variation (CNV), single-nucleotide variation (SNV), integrated viral sequences, and fusion genes, along with a correlation with the tumor stage for better disease prognosis [27,33,34].

In this study, we analyzed cfDNA from 30 patients diagnosed with oral squamous cell carcinoma (OSCC) and identified a statistically significant positive Pearson’s correlation (at 5% false discovery rate (FDR) between the cfDNA concentration and tumor stage, the emergence of secondary malignancies, and the lesion size. Additionally, we observed that elevated cfDNA levels were associated with increased recurrence rates and reduced survival probability in OSCC patients. Our genomic analyses revealed frequent CNVs in cfDNA at loci containing key cancer-related genes, including *TP53*, *TP63*, *FGFR1*, and *MLH1*. We have identified SNVs like insertions, deletions, and frameshift mutations in our cohort. Common SNVs identified in our samples were KMT2C, MUC3A, and MUC6, which have been previously reported in other cohort. Furthermore, the presence of oncogenic viral sequences, such as HPV, HSV, simian virus (SV), and enterovirus (EV), were detected in several patient samples, suggesting a potential role for viral integration in the oncogenic progression of OSCC. Notably, we identified a novel fusion gene, *TRMO-TRNT1*, representing a chimera of two genes involved in tRNA processing. *TRMO* (located on chromosome 9q22) encodes a tRNA methyltransferase, while *TRNT1* (on chromosome 3p26) is responsible for the addition of CCA nucleotides to tRNA molecules. According to the Genotype-Expression project (GTEx) data, both genes are ubiquitously expressed across human tissues [35]. Although *TRMO* has previously been linked to thyroid cancer, its role in OSCC has yet to be elucidated [36]. *TRNT1*, while not currently associated with cancer, is known to harbor mutations that cause sideroblastic anemia [37]. To validate our findings, we analyzed 199 publicly available RNA-seq datasets from OSCC patients and 192 from healthy controls. We observed *TRNT1-TRMO* in the OSCC data while this chimera was not observed in corresponding normal tissues. We further observed approximately 25% of the chimeric transcripts identified in our OSCC liquid biopsies were also detected in public OSCC tissue biopsy datasets and absent in healthy controls. Notably, ten chimeras exhibited expression levels as high as 50 reads per million underscoring their potential relevance. Collectively, our findings contribute to a more comprehensive understanding of the molecular landscape of OSCC. The identification of recurrent cfDNA alterations, viral integrations, and novel chimeric transcripts may support the development of minimally invasive biomarkers for early diagnosis, prognostication, and personalized therapeutic strategies in OSCC.

## 2. Materials and Methods

### 2.1. Methodology for Sample Size Collection

The sample size in our study was determined based on the availability of clinically well-characterized OSCC patients and matched healthy controls within the approved timeframe and ethical guidelines. A total of 30 OSCC patients were enrolled, representing diverse tumor stages and anatomical sub-sites, along with 25 healthy individuals serving as the controls. All participants provided informed consent, and the study was conducted under the approval of the relevant institutional ethics committee (Helsinki Committee approval-OSCC #0813-16-RMC). The inclusion criteria ensured representation across primary and secondary malignancies, and sample collection was standardized to minimize technical variabilities. This sample size allowed for preliminary statistical analyses and correlation assessments between cfDNA levels and clinical parameters, and expansion in future cohorts for higher statistical power.

### 2.2. Sample Collection

Blood samples of 30 OSCC patients were drawn before surgery and collected in cfDNA preservation tubes at the Rabin Medical Center. The blood samples were centrifuged at 300× *g* for 20 min at room temperature (RT) to separate the plasma and PBMCs. The plasma samples were then collected, aliquoted into Eppendorf tubes, and stored at −80 °C until use.

### 2.3. cfDNA Extraction

Aliquots (1 mL) of plasma from 30 patients were processed for cfDNA extraction using the Qiagen QIAamp ccfDNA/RNA Kit (Cat. No. 55184), Redwood, CA, USA in accordance with the manufacturer’s protocol and following informed patient consent. All procedures were conducted under the approval of the institutional ethics committee.

Inclusion Criteria:Patients diagnosed with histopathologically confirmed OSCC;Age ≥ 18 years;Availability of complete clinical and demographic data;Ability to provide informed consent;No prior systemic cancer therapy (chemotherapy or radiotherapy) at the time of sample collection.

Exclusion Criteria:Patients with other concurrent malignancies;A history of autoimmune or chronic inflammatory diseases;Recent major surgical procedures or trauma within the past month;An inadequate sample quality or volume for cfDNA analysis.

Appropriate buffers were added at the specified volume per 1 mL of serum and vortexed and incubated as instructed. Centrifugation was performed at 12,000× *g* for 3 min to pellet the precipitate. The supernatant was then transferred to a new tube, and ice-cold isopropanol was added in a 1:1 ratio, briefly vortexed, and centrifuged at 5000× *g* for 1 min. The washing step was conducted at 5000× *g* for 5 min. cfDNA was eluted in 50 µL of elution buffer.

Extracted cfDNA concentrations were measured using the Qubit dsDNA High Sensitivity Assay (Thermo Fisher Scientific, Waltham, MA, USA) with a Qubit 2.0 fluorometer. Additionally, a Bioanalyzer 2100 DNA High Sensitivity Assay (Agilent Technologies, Santa Clara, CA, USA) was performed to determine the cfDNA fragment size distribution.

### 2.4. Copy Number Analyses

Extracted cfDNA samples from all patients were sequenced using Illumina high-throughput whole genome sequencing (at least 50 million reads per sample, NextSeq 550). Paired-end sequencing was performed. For copy number analysis, the iChorCNA [38] and Gistic2.0 [39] pipelines were used with default parameters.

### 2.5. Chimera Identification and Validation

Using the *in-house* chimera detection pipeline ChiTaH [40], novel fusion genes were identified in sequenced cfDNA samples. To validate the ChiTaH results, the primers were designed upstream and downstream of the gene–gene junction sequence of a fusion gene (See Appendix A). PCR was performed using the Hylab Hy-Taq Ready Mix (2x) (Cat no. EZ3006/7/636) (For the PCR conditions, refer to Appendix A). PCR products were purified using a Macherey-Nagel NucleoSpin Gel and PCR Clean-up Kit (Cat no: 12303368), followed by Sanger sequencing done at Macrogen, Israel.

### 2.6. Gene Interaction Network Analysis

GeneMania [41] was used to find the gene regulators of the TRNT1 and TRMO genes. Only co-expressed and physical interaction partners were used for the study.

### 2.7. Identification of Coding Potential

The coding potential of a chimera was determined using the CPAT [42] and CNIT [43] web server tools, employing the default parameters.

### 2.8. Virus Detection in cfDNA

The VirusFinder2.0 [44] pipeline was employed to detect the presence of oncogenic viruses in circulating cfDNA samples using its default parameters This tool integrates sequence alignment and viral–host fusion detection to accurately identify the viral integration events, enabling the characterization of viral contributions to tumorigenesis. Its application in this study supports a non-invasive approach to uncovering virus-associated cancers and complements genomic analyses aimed at identifying the molecular drivers of the disease.

### 2.9. SNV Analysis

SNV analysis was performed using three tools. VarScanv2.3.9 [45] uses default parameters for analysis. Only exonic SNVs were considered for the study. The SNVs from VarScan were annotated using ANNOVAR (https://annovar.openbioinformatics.org/en/latest/user-guide/download/, accessed on 1 March 2025) [46]. DeepVariantv1.0 [47], and Mutect2 v4.1.4.1 [48]. The SNVs that were identified by all three tools were selected for the study.

### 2.10. Gene Ontology Analysis

Gene ontology analyses were performed using PantherDBv16 [49] software.

### 2.11. Statistical Analysis

Statistical analysis was performed using GraphPad Prism 10.4.2. A Shapiro–Wilk test was used for various correlation analyses. The t-test was performed to analyze the significant difference between the cfDNA concentration of healthy and OSCC patients. Survival analysis was conducted using a Kaplan–Meyer curve, which was included in the Prism GraphPadv10.4.2 software. A *p*-value was considered at ≤0.05 for statistical significance at at 5% FDR.

## 3. Results

It is known that high concentrations of cfDNA are observed in tissue injuries, cancer, and various autoimmune diseases. The demography of all 30 OSCC patients and 25 healthy individuals was considered in our studies (see Appendix A). The demographic parameters of all 30 OSCC patients with a clinical history were mentioned. (see Appendix A). The cfDNA samples of OSCC and healthy individuals were extracted and subjected to Bioanalyzer^TM^ (Agilent Technologies, Santa Clara, CA, USA), evaluations (Appendix A). Firstly, we studied the difference in the cfDNA concentrations of OSCC patients vs. healthy individuals and identified that the cfDNA in OSCC patients was significantly higher than the cfDNA concentrations of healthy individuals (Figure 1A) (*p*-value < *0.05*, *t*-test), thereby suggesting that OSCC patients have a significantly higher cfDNA concentration than healthy individuals. Next, we sought to understand the correlation of increased cfDNA levels with various tumorigenic parameters, like the tumor stage, a second malignancy, patient survival, and more. In this study, cancers of the oral cavity sub-site samples were considered. These included twenty tongues, eight alveolar ridges, one floor of the mouth, and one retromolar trigone. Eleven patients showed mandibular involvement at presentation, meaning the bone of the mandible was invaded by the tumor at diagnosis; nine had undergone segmental mandibulectomy, and two had experienced marginal mandibulectomy procedures. To check if increased cfDNA levels had a direct correlation with the tumor stage, we performed Pearson’s correlation analyses. We observed that even though the average cfDNA concentrations in patients with a tumor stage 3 or 4 were high, as compared to patients with tumor stages 1 and 2, the overall distributions were similar, other than a few cases with stage 3 or 4 tumors that presented very high concentrations of cfDNA in their blood samples (Figure 1B). These results indicate that cfDNA concentrations specify the presence of stage 3 or 4 OSCC tumors at *p*-value < 0.05.

We also observed similar results with secondary vs. primary malignancies and observed that patients with secondary malignancies had a high cfDNA concentration at a *p*-value < 0.05 (Figure 1C). Furthermore, the increased cfDNA concentration was significantly associated with a higher risk of disease recurrence in OSCC patients (Figure 1D). Then, using the Kaplan–Meyer survival analysis method on data describing the OSCC recurrence and death vs. cfDNA concentration, we found that patients with cfDNA concentrations of 27 ng/mL or less had a higher survival probability (68%; Figure 1E,F). These results indicate that cfDNA concentrations above a threshold of 27 ng/mL can be used to recognize tumor stages 3 and 4, which, furthermore, can be associated with decreased survival. This study thus helped in identifying the probability of survival, underlining the tumor stages using the cfDNA concentration as a biomarker. In the future, this method could be used as an even more sensitive biomarker for the early detection of cancer.

### 3.1. Oncogenic Viruses Were Identified When Analyzing cfDNA

In the present study, we detected partial contigs of HPV and other oncogenic viruses. Specifically, nine of thirty OSCC patients showed the presence of viral DNA integration into cfDNA (Figure 2A). Moreover, we observed sequences from HSV strains 1, 3, and 5, SV, HPV, and EV (Figure 2A). HSV integrations were identified in six OSCC patients in the present study; three patients had HSV-1, and three patients had HSV-5 integrations. The relation of EV with cancer has not been completely established; however, some evidence shows a link to malignancy in gliomas and to the progression of cervical cancers. SV has been identified in the serum of lung cancer patients. However, SV and EV have not yet been recognized as causing OSCC. These results indicate that our study revealed the presence of more than two viruses in OSCC patient sera (Figure 2B). However, the status of these viruses cannot yet be statistically correlated with the tumor stage in our cohort. Nonetheless, the presence of viruses may serve some oncogenic role in OSCC, and more samples should be used to identify their characteristics in OSCC patients.

### 3.2. A Novel Chimera Identified in cfDNA

Fusion genes that are produced by chromosomal translocations are common in lymphomas and leukemias yet are extremely rare in HNSCC [27,34]. Using our *in-house* chimera analysis tool, ChiTaH, we identified the novel fusion *TRMO*-*TRNT1* in the cfDNA samples of two patients (sequence, Appendix A-Fusion_OSCC_30). TRMO (tRNA methyl-O-transferase) and TRNT1 (tRNA nucleotide transferase I) were highly expressed in all tissue types. To validate this chimera at both RNA and DNA levels, we designed primers and performed PCR on the total RNA and DNA from OSCC biopsies and used patient 3 and patient 5 as positive controls (Figure 3). Subsequently, we conducted PCR on all 30 cfDNA samples and observed a 108 bp fused sequence in seven of them (Appendix A). The band containing this fusion was excised from the agarose gel, purified, and then assessed by Sanger sequencing, which confirmed the fusion junction of the two parental genes, *TRMO* and *TRN1* (Appendix A). This fusion was identified in seven patients using PCR against the junction sequence and the Sanger sequencing result, confirming the same, showing that the tumor has advanced malignancy and proliferation status. The altered expression of *TRMO* has been observed in patients with thyroid carcinoma [36], while altered *TRNT1* expression was seen in colorectal cancer patients [37]. These results suggest that fusion genes could be used as biomarkers and explored in the search for additional OSCC patients, which may be identified by means of the deep sequencing of cfDNA samples. Finally, we checked the coding potential of the observed chimera to produce some functional protein. For this, we utilized three tools commonly used for analyzing the coding/non-coding potential of a given DNA sequence. The scores obtained by the CPAT and CNIT programs suggested that the fusion has a <0.3% coding probability. Therefore, the fusion does not have coding potential (Appendix A) but it still may serve as a biomarker of the OSCC progression through a highly abundant non-coding cfDNA fragment. Next, we checked for the common interactors of *TRNT1-TRMO* genes (Figure 3B). Three common interactors are identified as *THADA*, *COMMD2*, and *CTBP2.* In addition, we downloaded 199 OSCC and 192 normal RNA-seq datasets from public databases (Appendix A Fusions in 192 OSCC patients from public dataset) and considered only those chimeras expressed in tumors while absent in healthy controls for our study. We found that 4 out of 16 chimeras identified in the OSCC plasma samples were also observed in the public biopsies of OSCC patients at 50 RPKM. We observed the presence of *TRNT1-TRMO* fusion in few OSCC patients from public dataset at a frequency of 5%. Our findings may help to further characterize novel abundant chimeras as biomarkers in OSCC for the early diagnostics and survival prognostics.

### 3.3. Copy Number Variation (CNV) Analysis of OSCC Patient cfDNA

CNV is commonly observed in many cancers. Various studies have identified CNV and somatic mutations using cfDNA [27,33,50,51,52]. Our aim here was to identify significant changes in the CNVs for OSCC patients using cfDNA genomic data. We performed CNV analysis on the sequenced cfDNA of OSCC patients using Gistic2.0, which assesses low pass copy number variations. The Q-value was set at 0.01 for the significance level. We observed significant amplifications in chromosomes 3q22.2, 7q35, and 15q26. Additional amplifications were also seen in chromosomes 8p21.3, 10q26,2, 2p25, and 1q21.3 (Figure 4A) (Appendix A OSCC_Copy number variations). Some of our findings correlate with the TCGA HNSCC data for CNV and significantly mutated genes. Next, we performed functional enrichment analysis using PantherDB and observed that commonly amplified genes in the present study were predicted to play a significant role in the GPCR pathways, opioid pathways, immune system activation pathways, apoptosis, and angiogenesis activation pathways (Figure 4C). Particularly, we identified deleterious regions in patient cfDNA samples. In our cohort, we observed that chromosomes 1q23, 2q22, 3p21, 3q26, 7q31, 8p21, 9q34, 13q13, and 14q21 had significant deletions (Figure 4B). Significantly mutated genes, like *TP53*, *TP63*, *FGFR1*, or *MLH1*, have been shown to present deletions in those regions and were also identified in the present study (Appendix A) [27,53,54,55]. We also found deleted regions of *THADA* in our cohort. Amplification of regions of the *CTBP2* genes has been observed in the present study. Both *THADA* and *CTBP2* are common interactors of fusion genes *TRNT1-TRMO* (Figure 3B). Our gene ontology (GO) analysis of amplification regions found that the deleterious genes in our samples assigned to the affected genes were mostly involved in immune activation pathways, apoptosis signaling pathways (particularly the *CCKR* gene), cell cycle pathways, and various metabolic pathways (Figure 4D). These results indicate that cfDNA analyses may reveal significant CNV in the OSCC samples and may be used for OSCC’s early diagnosis and prognosis.

### 3.4. SNV Analysis of OSCC Patients’ cfDNA

SNV analysis suggests structural variants and is key to identifying the mutations that cause cancer [53]. We used three tools to perform the SNV analysis of the cfDNA of OSCC patients (Appendix A Somatic nucleotide variations, annotation and filtering). We identified common SNVs present in our samples, such as *MUC6* (27%), *MUC3A* (70%), *FRG2C* (80%), *KMT2C* (26%), *AP2A2* (23%), *BAGE (*17%*)*, and *CCDC200 (*20%*)* as well as *TAF11L4*, *TAF11L5*, and *TAF11L6* (23% each) in our cohort, (Figure 5). In the present study, we considered non-synonymous mutations, frameshifts, and in-Del mutations in exonic regions. Most mutations observed were C > T, C > A, G > T, G > A. Some deletions in *MUC3A* and *FRGC2* have been observed. These observations are in concordance with other studies reported for HNSCC. These mutations can be used to diagnose OSCC.

Approximately 4% of the SNVs found had deletions in *KMT2C* and *MUC3A*, 1% were frameshift mutations observed in *MUC6*, and 20% were non-synonymous substitution mutations observed mainly in *KMT2C*, *MUC6*, *MUC3A*, *ZNF717*, and *SLC41A2* (Table 1). These mutations have been identified as significant mutations across various studies in HNSCC and OSCC. The mutations in these genes have been identified to play oncogenic roles in HNSCC.

## 4. Discussion

cfDNA has been explored as a prognostic and diagnostic biomarker in many cancers [11,12,14,16,18,24,56,57,58]. A correlation of cfDNA levels with various tumors has been established in glioblastoma and neuroblastoma. High cfDNA concentrations were observed in samples with cancer, as compared to the cfDNA from healthy samples [17,34]. Many studies have found that the concentration of cfDNA was correlated with tumor burden [19,59]. In the present study, we observed that the cfDNA concentration of OSCC patients was significantly higher than the cfDNA concentration of healthy patients. This result suggests that the cfDNA concentration of OSCC patients could be used for early and late diagnostics. Then, we explored the correlation between the cfDNA concentration from 30 OSCC patients and tumor stages. High cfDNA levels were observed in patients with stage 3 or 4 tumors, while comparatively lower cfDNA concentrations were observed in patients with stage 1 or 2 tumors. We also observed a direct correlation between a higher cfDNA concentration at presentation and the possibility of future recurrence and a secondary malignancy at a 5% FDR. We also observed that the threshold of a maximum cfDNA concentration for prognostication was <27 ng/mL. The Kaplan–Meyer survival analysis showed that the probability of survival was 68% for patients with OSCC at cfDNA concentrations above 27 ng/mL. Therefore, our study suggests that cfDNA could be used as a non-invasive method to detect OSCC at an early stage. Still, the analysis of more samples could have resulted in a better level of significance for the correlation between the cfDNA concentration and the tumor parameters.

Various genomic alterations in the cfDNA of OSCC patients have also been addressed. One such alteration was observed as the presence of viral DNA integrations. Deep sequencing of the cfDNA of OSCC patients revealed the presence of HPV, SV, HSV, and EV integrations. While a high proportion of patients (48%) did not show the presence of any virus, a few patients presented more than one virus. Interestingly, HPV has been studied as a cause of HNSCC, cervical cancer, and other gynecological and urological cancers [60,61,62,63,64,65,66,67,68,69]. We identified HPV in six out of thirty samples. HSV strains 1 and 5 were identified in six of thirty samples. HSV was previously reported in samples from HNSCC and OSCC patients, and its association with the progression of cancer was claimed [30,31,32]. Ev has been associated with malignant glioma [70]. SV was reported in samples of osteosarcoma and prostate cancer [71,72,73]. However, these viruses have not yet been associated with OSCC. Future studies could establish the association of these viruses with OSCC and determine if they are involved in oncogenic progression.

Using the *in-house* ChiTaH pipeline we identified the novel fusion *TRMO*-*TRNT1*. PCR and Sanger sequencing validated the presence of this fusion in seven samples from patients with late-stage tumors. A mutation in *TRMO* has been associated with thyroid carcinoma [36]. *TRNT1* has not been associated with any cancer thus far, although its mutation is responsible for causing congenital disease and sideroblastic anemia [37]. The fusion product does not have any coding potential, according to CPAT [42] and CNIT [43]. Gene regulatory studies revealed three potential regulators of both TRNT1 and TRMO: THADA, CTBP2, and COMMD2. This fusion was observed in RNA-seq of OSCC tissue biopsy and is absent in healthy controls. Apart from *TRNT1-TRMO*, three other fusions were identified in cfDNA of our cohort that were observed in RNA seq of public OSCC tissue biopsy samples. Future research could explore the mechanism of action of this fusion gene in OSCC development.

CNV in tissues, as well as in cfDNA, has been studied and has been shown to be associated with many cancers, including OSCC [33,50]. In the present study, we found regions of chromosomes 3q22.2, 7q35, and 15q26 to be amplified. Amplifications were also seen in chromosomes 8p21.3, 10q26,2, 2p25, and 1q21.3. Gene ontology pathway analysis of the genes at these loci revealed them to be associated with angiogenesis, immune activation pathways, and GPCR, as well as interleukin, VEGF, FGF, and vasopressin activation pathways. The deleterious regions at chromosomes 1q23, 2q22, 3p21, 3q26, 7q31, 8p21, 9q34, 13q13, and 14q21 affect the genes associated with apoptosis, *CCKR*, the cell cycle, and immune activation and cell signaling pathways. Mutations in these genes and the structural variations associated with these loci have been studied in TCGA and by other groups. Our study agrees with TCGA and previous studies [27,54,55,74,75]. We also identified deleted regions of *THADA* in our cohort. Amplification of regions of the *CTBP2* gene was observed in the present study. Both *THADA* and *CTBP2* are common regulators of the fusion gene *TRNT1-TRMO*. In the future, it will be interesting to investigate whether structural variations in *THADA*, *COMMD2*, and *CTBP2* play a significant role in OSCC progression and whether the fusion gene *TRNT1-TRMO* is involved in CNV. Additionally, targeted sequencing of these genes or CNV analysis of these chromosomal loci could be used as potential biomarkers for OSCC. Additionally, from the 199 public OSCC RNA-seq datasets, we found that 50% of the chimeras identified in the OSCC liquid biopsies were also observed in public biopsies. Furthermore, 10 chimeras exhibited abundances as high as 88 RPKM and may serve as potential biomarkers for OSCC. These findings could help explore their association with cancer and enhance applications in early diagnostics and survival prognostics.

SNV analysis helps in understanding the key mutations leading to oncogenesis [27,74,76,77,78]. They help in identifying potential risk mutations that can lead to carcinogenesis. SNVs can be a result of various factors, including environmental factors, smoking status, chewing betel quid, and viral integration status [67,79,80,81]. SNV analysis was performed in the present study to identify significant mutations in genes. Our study suggested common mutations in genes such as *MUC6* (27%), *MUC3A* (70%), *FRG2C* (80%), *KMT2C* (26%), *AP2A2* (23%), *BAGE (*17%*)*, and *CCDC200 (*20%*)* in our cohort. Mutational profiles of *TAF11L4*, *TAF11L5*, and *TAF11L6* (23% each) have not been reported in HNSCC. Previous studies reported that the Mucin genes were altered in OSCC. Most mutated Mucin genes are *MUC1*, *MUC4*, *MUC3*, and *MUC6* [82,83,84]. *MUC1* and *MUC4* have been identified as prognostic biomarkers for OSCC, which leads to oral mucosa carcinogenesis [82,83,84,85]. *MUC6* has been recently identified as a potential biomarker in Asian OSCC patients. The mutations in *MUC6* have been associated with malignancy and tumorigenesis of oral carcinoma [84,85,86]. Recently, *MUC3A* has also been associated with the progression of oral cancer. Mutations in *MUC3A* have been linked to oncogenesis [84,85,86,87]. Another study showed that *ZNF717* and *MUC3A* are mutated in tongue squamous cell carcinoma [87]. In the present study, we identified the SNV of *ZNF717* in one sample. *FRG2C* is a signature gene mutated in ESCC (esophageal squamous cell carcinoma) and proliferative verrucous leukoplakia (PVL) [88,89]. Another gene highly mutated in the present study cohort, *KMT2C*, was observed to be rarely mutated in South Asian patients with HNSCC [89,90]. *BAGE* was identified as a clinically significant immunogenic antigen in HNSCC patients [91]. *AP2A2*, an adapting protein family molecule, has not been reported to be oncogenic. However, it has been linked to inhibiting the activity of Erlotinib, an anti-cancer drug for lung cancer [92].

All these mutational signatures of genes can be drug targets for precision medicines. Also, these genes can be used further to study the relevance of mutation in OSCC progression and thereby can be used as a biomarker for non-invasive cancer diagnosis and prognosis.

## 5. Conclusions

Our study highlights the potential of circulating cfDNA as a non-invasive biomarker for the diagnosis, prognosis, and therapeutic monitoring of OSCC. Elevated cfDNA levels correlated significantly with the tumor stage and patient outcomes, suggesting its value in early detection and recurrence risk assessment. We also uncovered a spectrum of genomic alterations in cfDNA, including viral integrations, chimeric transcripts, copy number variations (CNVs), and single-nucleotide variants (SNVs), many of which have known or emerging relevance to OSCC progression. Notably, we identified and validated a novel fusion gene, TRMO-TRNT1, which is associated with late-stage tumors, and observed recurring mutations in genes such as MUC3A, MUC6, FRG2C, and KMT2C, with implications for oncogenesis and potential as therapeutic targets. CNV analysis revealed altered regions linked to the immune response and oncogenic pathways, supporting the use of these alterations as molecular biomarkers. The consistency of several chimeric transcripts with public datasets further underscores their potential clinical relevance. The limitation of the present study is the low number of samples studied. Altogether, our findings establish a robust foundation for integrating cfDNA-based assays into OSCC diagnostics and precision medicine frameworks while pointing toward novel molecular targets for future therapeutic exploration. Continued research with larger cohorts and functional validation will be essential to fully realize the translational potential of these biomarkers.

## Figures and Tables

**Figure 1 cancers-17-01901-f001:**
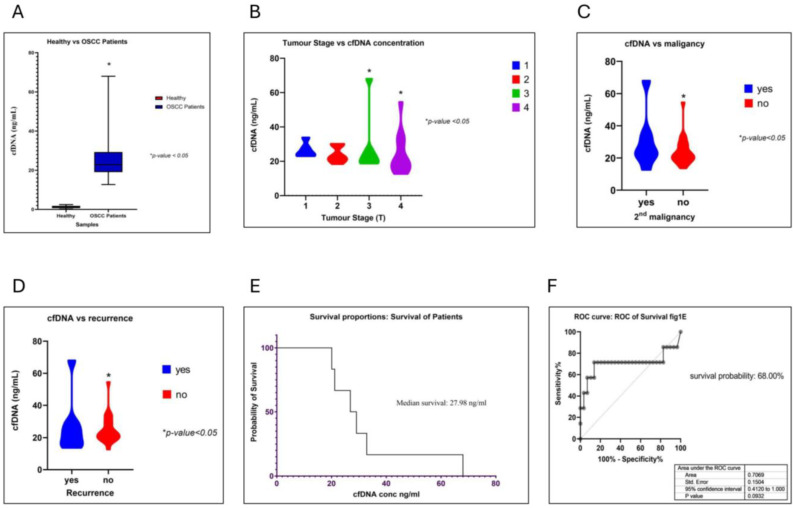
Correlation of cfDNA concentration with different tumor profiles. (**A**) cfDNA concentration of healthy vs. OSCC patients (*p*-value ≤ 0.05) using *t*-test. (**B**) cfDNA concentration compared with tumor stage using Shapiro–Wilk normality. (**C**) cfDNA vs. secondary malignancy. (**D**) cfDNA vs. recurrence. (**E**) The probability of survival was determined using a survival curve, with the threshold of survival being 27 ng/mL. (**F**) Survival analysis using the Kaplan–Meyer method showed a 68% probability of survival within the cohort. * Indicates statistical significance at *p*-value < 0.05.

**Figure 2 cancers-17-01901-f002:**
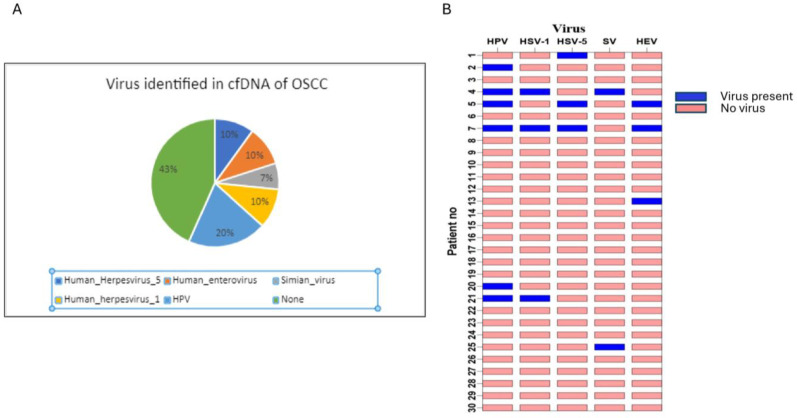
Identification of viral DNA in cfDNA samples. (**A**) Pie chart showing the percentage of samples containing viral DNA. (**B**) Presence of different viral DNA contigs in the samples. Some patients had DNA from more than one virus in their sample.

**Figure 3 cancers-17-01901-f003:**
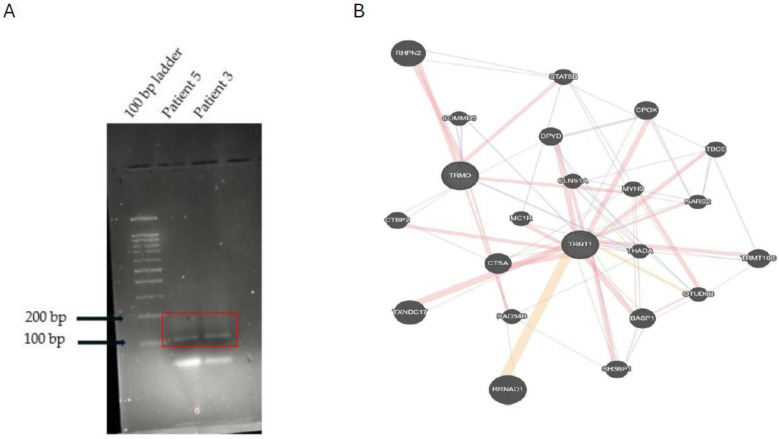
Identification of a novel fusion. (**A**) Agarose gel of PCR products from two confirmed patient samples with the fusion highlighted by a red box. (**B**) Gene–gene interaction network analysis. The orange line represents the co-expressed genes, and the purple line represents physical interactions. *CTBP2*, *COMMD2*, and *THADA* are the common interactors of the parental genes.

**Figure 4 cancers-17-01901-f004:**
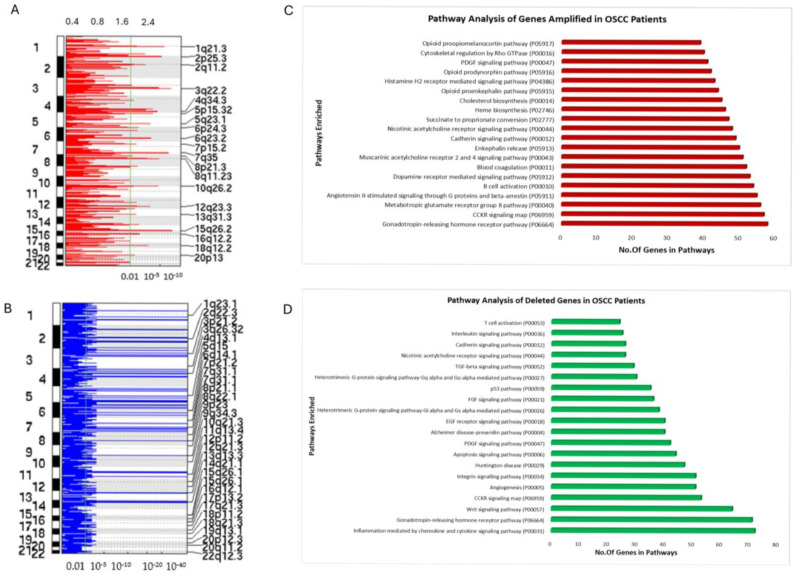
Copy number variation analyses of cfDNA sequences: CNV analysis was conducted on 30 patient cfDNA samples using Gistic2.0. An amplification plot suggests (**A**) amplification in the regions 8p21.3, 10q26,2, 2p25, and 1q21.3 and (**B**) deletions in chromosomes 1q23, 2q22, 3p21, 3q26, 7q31, 8p21, 9q34, 13q13, and 14q21. (**C**,**D**) Gene ontology analysis of amplified (**C**) and deleted regions (**D**) using PantherDB. Genes related to various signaling pathways and the apoptosis pathway were enriched.

**Figure 5 cancers-17-01901-f005:**
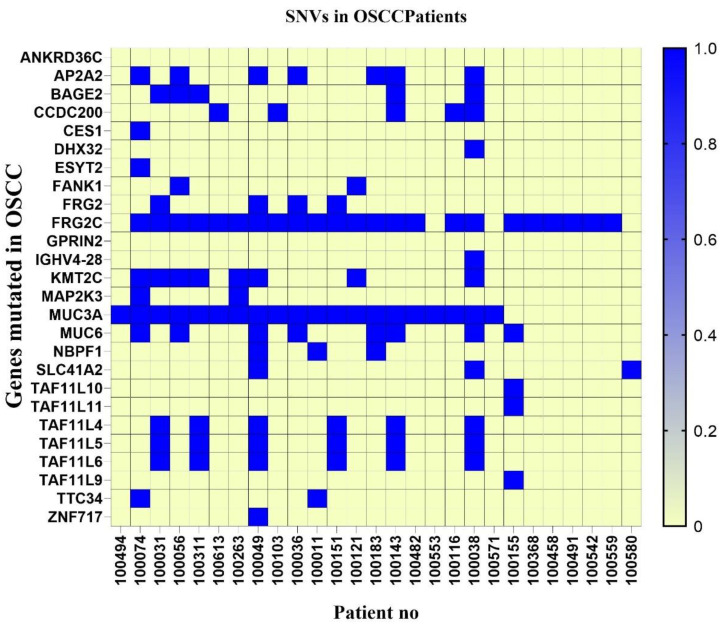
SNV analysis of OSCC patients. The heatmap shows SNVs identified in our OSCC cohort. Blue indicates SNVs in the sample, and yellow indicates no SNVs identified. On the Y-axis are genes that are mutated in our cohort. *MUC3A*, *MUC6*, *FRGC2*, *TAF11*, and *KMT2C* have high mutation rates in patients’ samples.

**Table 1 cancers-17-01901-t001:** SNVs in OSCC samples. The frequencies of non-synonymous mutations in OSCC patients’ cfDNA.

Mutations	Observed Frequencies
Deletions	4%
Non-synonymous	20%
Frameshift	1%
Insertion	0.1%
Transversion	1.2%

## Data Availability

All data are available upon publication.

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
