# Peer review of "Applications for Circulating Cell-Free DNA in Oral Squamous Cell Carcinoma: A Non-Invasive Approach for Detecting Structural Variants, Fusions, and Oncoviruses"

_cancers, 2025, doi:10.3390/cancers17121901_

Round 1
Reviewer 1 Report
Comments and Suggestions for Authors
Comments to author
1). introduction section need to improve following article cite: https://doi.org/10.1016/j.oor.2024.100614
2). Materials and methods
Virus detection in cfDNA to briefly write materials and methods
3). Fig. 3A to provide high clarity images
4). Discussion need to improve to cite related article.
5). Author must be provide conclusion section with key outcomes of results and future recommendation or suggesiton.
Author Response
Comments to author
1). introduction section need to improve following article cite: https://doi.org/10.1016/j.oor.2024.100614
>ADDED.
2). Materials and methods
Virus detection in cfDNA to briefly write materials and methods
>We extended this part thanks to the reviewer.
3). Fig. 3A to provide high clarity images
>We improved the image. To note that it is the gel photo with a high resolution.
4). Discussion need to improve to cite related article.
>We added the article.
5). Author must be provide conclusion section with key outcomes of results and future recommendation or suggestion.
>Added thanks to the reviewer.
Reviewer 2 Report
Comments and Suggestions for Authors
The manuscript is about Circulating cell-free DNA and its relationship with OSCC. The topic is relevant and current. The abstract contains pertinent information. The introduction section is concise, well written and the objectives are well defined. The Materials And Methods section could include inclusion and exclusion criteria and ethical issues. The results and discussion sections are very well written. The manuscript presents promising results for non-invasive tests to monitor patients with OSCC.
Author Response
The manuscript is about Circulating cell-free DNA and its relationship with OSCC. The topic is relevant and current. The abstract contains pertinent information. The introduction section is concise, well written and the objectives are well defined. The Materials And Methods section could include inclusion and exclusion criteria and ethical issues. The results and discussion sections are very well written. The manuscript presents promising results for non-invasive tests to monitor patients with OSCC.
We thank the reviewer for their support!
Reviewer 3 Report
Comments and Suggestions for Authors
1.The authors need to restructure their introduction section with emphasis on relevance on available scientific evidence specific to their study instead of adding part of their results & methodology in the introduction section.
2.There is no mention of the methodology used for sample size estimation in the materials & methods section.
3.There is no mention whether informed consent was taken for the patients in the materials & methods section.
4.An elaborate inclusion & exclusion criteria needs to be added in the materials & methods section.
5.The authors need to add a note on the limitations & the future implications based on the results of the present study in the discussion section before conclusion.
6.The authors need to elaborate the conclusion section based on their views/suggestions and results of their present study
7.Minor written english language correction is required in the manuscript.
8.Further comments are highlighted in the attached manuscript.

1.The authors need to restructure their introduction section with emphasis on relevance on available scientific evidence specific to their study instead of adding part of their results & methodology in the introduction section.
2.There is no mention of the methodology used for sample size estimation in the materials & methods section.
3.There is no mention whether informed consent was taken for the patients in the materials & methods section.
4.An elaborate inclusion & exclusion criteria needs to be added in the materials & methods section.
5.The authors need to add a note on the limitations & the future implications based on the results of the present study in the discussion section before conclusion.
6.The authors need to elaborate the conclusion section based on their views/suggestions and results of their present study
7.Minor written english language correction is required in the manuscript.
8.Further comments are highlighted in the attached manuscript.
Author Response
1.The authors need to restructure their introduction section with emphasis on relevance on available scientific evidence specific to their study instead of adding part of their results & methodology in the introduction section.
>We improved the Introduction.
2.There is no mention of the methodology used for sample size estimation in the materials & methods section.
>We added the Methodology of the sample size, the informed consent part and the inclusion/exclusion parts.
3.There is no mention whether informed consent was taken for the patients in the materials & methods section.
>We added the informed consent part and the inclusion/exclusion criteria.
4.An elaborate inclusion & exclusion criteria needs to be added in the materials & methods section.
>We added the inclusion/exclusion criteria.
5.The authors need to add a note on the limitations & the future implications based on the results of the present study in the discussion section before conclusion.
> We added it in the new Conclusion part.
6.The authors need to elaborate the conclusion section based on their views/suggestions and results of their present study
> We added the new Conclusion part.
7.Minor written English language correction is required in the manuscript.
>We improved the text and marked in red.
8.Further comments are highlighted in the attached manuscript.
>We thank the reviewer for their comments.

Reviewer 4 Report
Comments and Suggestions for Authors
Dear Authors,
this is an interesting paper with certain aspects that need to be addressed.
Abstract structure is missing: introduction, MM, results and conclusion.
I would like to see more info about liquid biopsy in the introduction section.
Figure 3 - if it possible to do a better resolution of a picture; it's a little bit blurry.
Results are not well presented; first sentence (line 14) are not suitable for results section but for discussion. Please, only state the obtained results and avoid interpreting them in the results section.
Also avoid in the entier text "in our study", "we identified"...etc... Instead of that use "in the present study"... "We have also addressed various genomic alterations in the cfDNA of OSCC patients" - Various genomic alterations in the cfDNA of OSCC patients have also been addressed.
Limitations of the study and conclusion are missing in the discussion section.
Reference list must be corrected; different styles are used.
Best regards
Author Response
Dear Authors,
this is an interesting paper with certain aspects that need to be addressed.
Abstract structure is missing: introduction, MM, results and conclusion.
>In accordance with the reviewer’s feedback, we have included a concluding section to summarize and contextualize our findings.
I would like to see more info about liquid biopsy in the introduction section.
>We improved the Liquid Biopsy part in the Introduction.
Figure 3 - if it possible to do a better resolution of a picture; it's a little bit blurry.
>We improved the resolution.
Results are not well presented; first sentence (line 14) are not suitable for results section but for discussion. Please, only state the obtained results and avoid interpreting them in the results section.
>We improved according to the reviewers' comments.
Also avoid in the entier text "in our study", "we identified"...etc... Instead of that use "in the present study"... "We have also addressed various genomic alterations in the cfDNA of OSCC patients" -
>We fixed all the places as mentioned.
Limitations of the study and conclusion are missing in the discussion section.
>We added the Conclusions part.
Reference list must be corrected; different styles are used.
>Fixed.
Round 2
Reviewer 4 Report
Comments and Suggestions for Authors
Dear Authors,
thank you for your revised version.
I have no additional comments.
Best regards